# Constructing Semantics-Aware Adversarial Examples with Probabilistic Perspective

## Abstract

In this study, we introduce a novel, probabilistic viewpoint on adversarial examples, achieved through box-constrained Langevin Monte Carlo (LMC). Proceeding from this perspective, we develop an innovative approach for generating semantics-aware adversarial examples in a principled manner. This methodology transcends the restriction imposed by geometric distance, instead opting for semantic constraints. Our approach empowers individuals to incorporate their personal comprehension of semantics into the model. Through human evaluation, we validate that our semantics-aware adversarial examples maintain their inherent meaning. Experimental findings on the MNIST and SVHN datasets demonstrate that our semantics-aware adversarial examples can effectively circumvent robust adversarial training methods tailored for traditional adversarial attacks.

## 1 Introduction

The purpose of generating adversarial examples is to deceive a classifier by making minimal changes to the original data's meaning. In image classification, most existing adversarial techniques ensure the preservation of adversarial example semantics by limiting their geometric distance from the original image [18, 6, 2, 12]. These methods are able to deceive classifiers with a very small geometric based perturbation. However, when targeting robust classifiers trained using adversarial methods, an attack involving a relatively large geometric distance may be necessary. Unfortunately, these considerable distances can be so vast that they ultimately undermine the original image's semantics, going against the core objective of creating adversarial examples. As illustrated in the left portion of Figure 1, when applying the PGD attack [12] constrained by $L_2$ norm on a robust classifier, the attacked images that successfully deceive the classifier consistently lose their original meaning, which is undesirable.

To counter this problem, we propose an innovative approach for generating semantics-aware adversarial examples. Instead of being limited by geometric distance, our approach hinges on a proposed semantic divergence. Specifically, we treat generating adversarial examples as a box-constrained non-convex optimization problem. We employ box-constrained Langevin Monte Carlo (LMC) to find near-optimal solutions for this complex problem. As LMC samples converge to a stationary distribution, we gain a probabilistic understanding of the adversarial attack. Within this probabilistic perspective, the geometric constraint of the adversarial attack can be viewed as a distribution. By replacing this geometric-based distribution with a semantic-based distribution, we can define a semantics-aware adversarial attack in a principled manner. The corresponding divergence induced by the semantic-based distribution is called semantic divergence. Our semantics-aware adversarial attack is capable of deceiving robust classifiers while preserving most of the original image's semantics, as demonstrated in the right section of Figure 1.

Submitted to 37th Conference on Neural Information Processing Systems (NeurIPS 2023). Do not distribute.

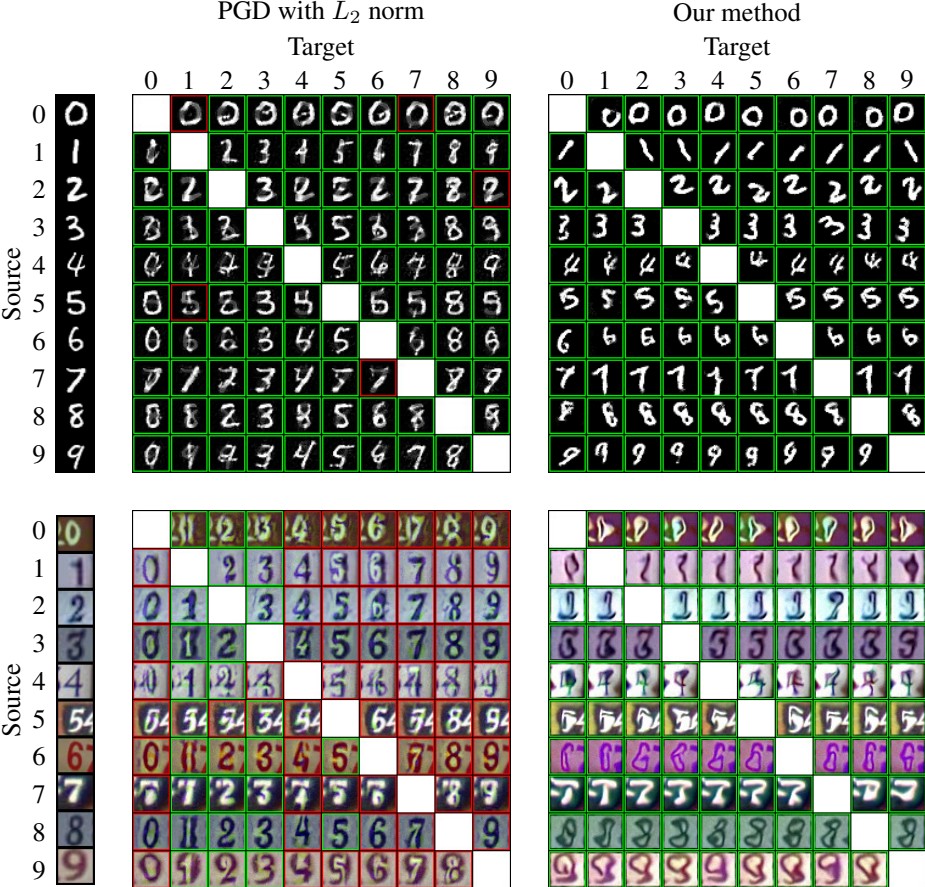

Figure 1: **Top left**: Targeted attack on an adversarially trained MadryNet [12] for MNIST using Projected Gradient Descent (PGD) with $L_2$ norm. To ensure successful targeted attacks in most cases, we increased the $\epsilon$ to 5. **Bottom left**: Targeted attack on an adversarially trained ResNet18 [8] for SVHN using PGD with $L_2$ norm and $\epsilon = 5$. **Top right & Bottom right**: Our proposed method applied to targeted attacks on the same MadryNet and ResNet18 for MNIST and SVHN, respectively. A green border signifies a successful deception of the victim classifier, while a red border indicates failure. **Notably, with PGD, a successful attack often results in the alteration of the source image's semantics, which is undesirable.** Additional PGD attack examples are provided in Appendix E.

## 2 Preliminaries

### 2.1 Adversarial examples

The notion of adversarial examples was first introduced by Szegedy et al. [18]. Let's assume we have a classifier $C : [0, 1]^n \to \mathcal{Y}$, where $n$ represents the dimension of the input space and $\mathcal{Y}$ denotes the label space. Given an image $\mathbf{x}_{\text{ori}} \in [0, 1]^n$ and a target label $y_{\text{tar}} \in \mathcal{Y}$, the optimization problem for finding an adversarial instance for $\mathbf{x}_{\text{ori}}$ can be formulated as follows:

$$\text{minimize } \mathcal{D}(\mathbf{x}_{\text{ori}}, \mathbf{x}_{\text{adv}}) \quad \text{such that } C(\mathbf{x}_{\text{adv}}) = y_{\text{tar}} \text{ and } \mathbf{x}_{\text{adv}} \in [0, 1]^n$$

Here, $\mathcal{D}$ is a distance metric employed to assess the difference between the original and perturbed images. This distance metric typically relies on geometric distance, which can be represented by $L_0$, $L_2$, or $L_\infty$ norms.

However, solving this problem is challenging. As a result, Szegedy et al. [18] propose a relaxation of the problem:

$$\text{minimize } \mathcal{L}(\mathbf{x}_{\text{adv}}, y_{\text{tar}}) := c_1 \cdot \mathcal{D}(\mathbf{x}_{\text{ori}}, \mathbf{x}_{\text{adv}}) + c_2 \cdot f(\mathbf{x}_{\text{adv}}, y_{\text{tar}}) \quad \text{such that } \mathbf{x}_{\text{adv}} \in [0, 1]^n \quad (1)$$

where $c_1$, $c_2$ are constants, and $f$ is an objective function closely tied to the classifier's prediction. For example, in [18], $f$ is the cross-entropy loss function, while Carlini and Wagner [2] suggest several different choices for $f$. Szegedy et al. [18] recommend solving (1) using box-constrained L-BFGS.

## 2.2 Adversarial training

Adversarial training, a widely acknowledged method for boosting adversarial robustness in deep learning models, has been extensively studied [18, 6, 10, 12]. This technique uses adversarial samples as (part of) the training data, originating from Szegedy et al. [18], and has evolved into numerous variations. In this paper, we apply the min-max problem formulation by Madry et al. [12] to determine neural network weights, denoted as $\theta$. They propose choosing $\theta$ to solve:

$$\min_{\theta} \mathbb{E}_{(\mathbf{x},y) \sim p_{\text{data}}} \left[ \max_{\|\delta\|_p \leq \epsilon} \mathcal{L}_{\text{CE}}(\theta, \mathbf{x} + \delta, y) \right] \tag{2}$$

where $p_{\text{data}}$ represents the data distribution, $\mathcal{L}_{\text{CE}}$ is the cross-entropy loss, $\|\cdot\|_p$ denotes the $L_p$ norm, and $\epsilon$ specifies the radius of the corresponding $L_p$ ball. In what follows, we will use the term "robust classifier" to refer to classifiers that have undergone adversarial training.

## 2.3 Energy-based models (EBMs)

An Energy-based Model (EBM) [9, 4] involves a non-linear regression function, represented by $E_\theta$, with a parameter $\theta$. This function is known as the energy function. Given a data point, $\mathbf{x}$, the probability density function (PDF) is given by:

$$p_\theta(\mathbf{x}) = \frac{\exp(-E_\theta(\mathbf{x}))}{Z_\theta} \tag{3}$$

where $Z_\theta = \int \exp(-E_\theta(\mathbf{x})) \mathrm{d}\mathbf{x}$ is the normalizing constant that ensures the PDF integrates to 1.

## 2.4 Langevin Monte Carlo (LMC)

Langevin Monte Carlo (also known as Langevin dynamics) is an iterative method that could be used to find near-minimal points of a non-convex function $g$ [13, 25, 20, 14]. It involves updating the function as follows:

$$\mathbf{x}_0 \sim p_0, \quad \mathbf{x}_{t+1} = \mathbf{x}_t - \frac{\epsilon^2}{2} \nabla_x g(\mathbf{x}_t) + \epsilon \mathbf{z}_t, \quad \mathbf{z}_t \sim \mathcal{N}(0, I) \tag{4}$$

where $p_0$ could be a uniform distribution. Under certain conditions on the drift coefficient $\nabla_x g$, it has been demonstrated that the distribution of $\mathbf{x}_t$ in (4) converges to its stationary distribution [3, 14], also referred to as the Gibbs distribution $p(\mathbf{x}) \propto \exp(g(\mathbf{x}))$. This distribution concentrates around the global minimum of $g$[5, 24, 14]. If we choose $g$ to be $-E_\theta$, then the stationary distribution corresponds exactly to the EBM's distribution defined in (3). As a result, we can draw samples from the EBM using LMC. By replacing the exact gradient with a stochastic gradient, we obtain Stochastic Gradient Langevin Dynamics (SGLD) [23, 19].

## 2.5 Training EBM

To train an EBM, we aim to minimize the minus expected log-likelihood of the data , represented by

$$\mathcal{L}_{\text{EBM}} = \mathbb{E}_{X \sim p_d}[-\log p_\theta(X)] = \mathbb{E}_{X \sim p_d}[E_\theta(X)] - \log Z_\theta$$

where $p_d$ is the data distribution. The gradient is

$$\nabla_\theta \mathcal{L}_{\text{EBM}} = \mathbb{E}_{X \sim p_d}[\nabla_\theta E_\theta(X)] - \nabla_\theta \log Z_\theta = \mathbb{E}_{X \sim p_d}[\nabla_\theta E_\theta(X)] - \mathbb{E}_{X \sim p_\theta}[\nabla_\theta E_\theta(X)] \tag{5}$$

(see [16] for derivation). The first term of $\nabla_\theta \mathcal{L}_{\text{EBM}}$ can be easily calculated as $p_d$ is the distribution of the training set. For the second term, we can use LMC to sample from $p_\theta$ [9].

Effective training of an energy-based model (EBM) typically requires the use of techniques such as sample buffering and regularization. For more information, refer to the work of Du and Mordatch [4].

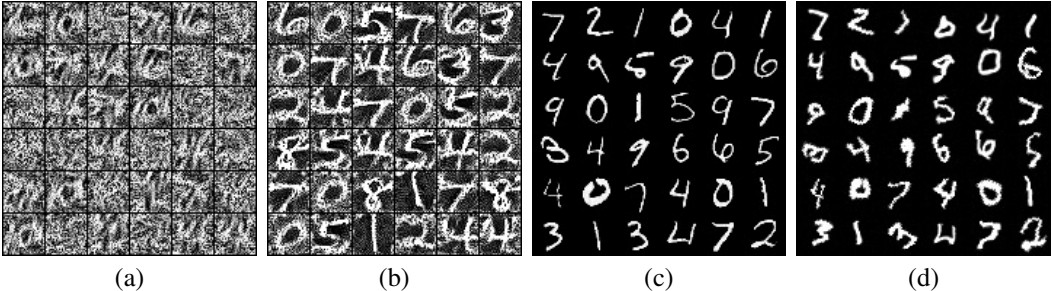

| (a) | (b) | (c) | (d) |

Figure 2: **(a)** and **(b)** display samples drawn from $p_{\text{vic}}(\cdot; y_{\text{tar}})$ with the victim classifier being non-adversarially trained and adversarially trained, respectively. **(c)** showcases samples from $p_{\text{dis}}(\cdot; \mathbf{x}_{\text{ori}})$ when $\mathcal{D}$ is the square of $L_2$ norm. **(d)** illustrates $t(\mathbf{x}_{\text{ori}})$ for $t \sim \mathcal{T}$, where $\mathcal{T}$ represents a distribution of transformations, including TPS (see Section 4.2), scaling, rotation, and cropping. The $\mathbf{x}_{\text{ori}}$s in (c) and (d) consist of the first 36 images from the MNIST test set.

## 3  Generating semantics-aware adversarial examples

In this section, we introduce a probabilistic approach to understanding adversarial examples. Through this lens, we establish the concept of semantic divergence, offering an alternative to conventional geometric distance. This concept of semantic divergence enables individuals to integrate their unique understanding of semantics into the model, thereby facilitating the creation of semantics-aware adversarial examples.

### 3.1  A probabilistic perspective on adversarial examples

LMC and SGLD are not directly applicable to the optimization problem presented in (1) due to their incompatibility with box-constrained optimization problems. To overcome this limitation, Lamperski [11] proposed Projected Stochastic Gradient Langevin Algorithms (PSGLA). By employing PSGLA to generate samples near the solution of the optimization problem specified in (1), we obtain the subsequent update rule:

$$\mathbf{x}_0 \sim p_0, \quad \mathbf{x}_{t+1} = \Pi_{[0,1]^n} \left( \mathbf{x}_t - \frac{\epsilon^2}{2} \nabla_x \mathcal{L}(\mathbf{x}_t, y_{\text{tar}}) + \epsilon \mathbf{z}_t \right), \quad \mathbf{z}_t \sim \mathcal{N}(0, I) \tag{6}$$

where $\Pi[0,1]^n$ is a clamp projection that enforces the constraints within the $[0,1]^n$ interval. We refer to the stationary distribution of PSGLA as the adversarial distribution $p_{\text{adv}}(\mathbf{x}; y_{\text{tar}}) \propto \exp(-\mathcal{L}(\mathbf{x}, y_{\text{tar}}))$, since samples drawn from this distribution are in close proximity to the optimal value of the optimization problem presented in (1).

Then by definition of $\mathcal{L}$, the adversarial distribution can be represented as a product of expert distributions [9]:

$$p_{\text{adv}}(\mathbf{x}_{\text{adv}}; \mathbf{x}_{\text{ori}}, y_{\text{tar}}) \propto p_{\text{vic}}(\mathbf{x}_{\text{adv}}; y_{\text{tar}}) p_{\text{dis}}(\mathbf{x}_{\text{adv}}; \mathbf{x}_{\text{ori}}) \tag{7}$$

where $p_{\text{vic}}(\mathbf{x}_{\text{adv}}; y_{\text{tar}}) \propto \exp(-c_2 \cdot f(\mathbf{x}_{\text{adv}}, y_{\text{tar}}))$ denote the victim distribution and $p_{\text{dis}}(\mathbf{x}_{\text{adv}}; \mathbf{x}_{\text{ori}}) \propto \exp(-c_1 \cdot \mathcal{D}(\mathbf{x}_{\text{ori}}, \mathbf{x}_{\text{adv}}))$ represent the distance distribution.

The victim distribution $p_{\text{vic}}$ is dependent on the victim classifier. As suggested by Szegedy et al. [18], $f$ could be the cross-entropy loss of the classifier. We can sample from this distribution using Langevin dynamics. Figure 2(a) presents samples drawn from $p_{\text{vic}}$ when the victim classifier is subjected to standard training, exhibiting somewhat indistinct shapes of the digits. This implies that the classifier has learned the semantics of the digits to a certain degree, but not thoroughly. In contrast, Figure 2(b) displays samples drawn from $p_{\text{vic}}$ when the victim classifier undergoes adversarial training. In this scenario, the shapes of the digits are clearly discernible. This observation suggests that we can obtain meaningful samples from adversarially trained classifiers, indicating that such classifiers depend more on semantics, which corresponds to the fact that an adversarially trained classifier is more difficult to attack. A similar observation concerning the generation of images from an adversarially trained classifier has been reported by Santurkar et al. [15].

The distance distribution $p_{\text{dis}}$ relies on $\mathcal{D}(\mathbf{x}_{\text{ori}}, \mathbf{x}_{\text{adv}})$, representing the distance between $\mathbf{x}_{\text{adv}}$ and $\mathbf{x}_{\text{ori}}$. By its nature, samples that are closer to $\mathbf{x}_{\text{ori}}$ may yield a higher $p_{\text{adv}}$, which is consistent with

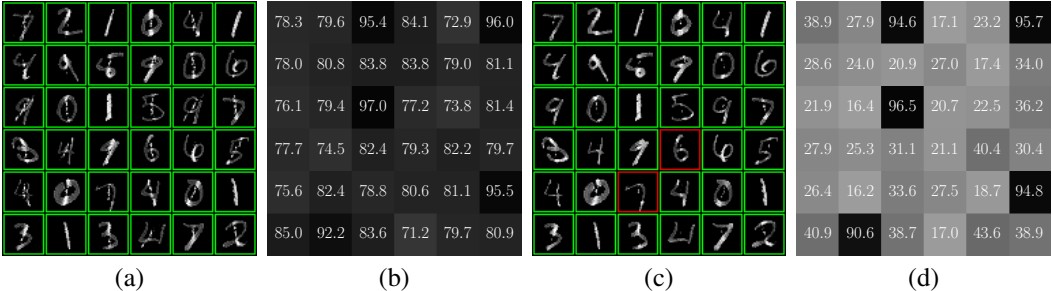



(a)            (b)            (c)            (d)



Figure 3: **(a)**: Samples from $p_{\text{adv}}(\cdot; \mathbf{x}_{\text{ori}}, y_{\text{tar}}) \propto \exp(-c_1 \cdot \mathcal{D}(\mathbf{x}_{\text{ori}}, \mathbf{x}_{\text{adv}})) \exp(-c_2 \cdot f(\mathbf{x}_{\text{adv}}, y_{\text{tar}}))$, where $\mathcal{D}$ is the $L_2$ norm, $f$ is the cross-entropy $f_{\text{CE}}$, $\mathbf{x}_{\text{ori}}$ are the first 36 images from the MNIST test set, $y_{\text{tar}}$ are set to 1, $c_1$ is $10^{-3}$, and $c_2$ is $10^{-2}$. **(c)**: Similar to (a), but with $f$ replaced by $f_{\text{CW}}$, as introduced in section 4.1. Essentially, this case applies the $L_2$ CW attack [2] using LMC instead of Adam optimization. A green border indicates successful deception of the victim classifier, while a red border signifies failure. **(b) & (d)**: the predictive probability (softmax probability) of the target class, corresponding to each digit of Figures (a) and (c) on a one-to-one basis.

the objective of generating adversarial samples. Moreover, if $\mathcal{D}$ represents the square of the $L_2$ norm, then $p_{\text{dis}}$ becomes a Gaussian distribution with a mean of $\mathbf{x}_{\text{ori}}$ and a variance determined by $c_1$. Figure 2(c) portrays samples drawn from $p_{\text{dis}}$ when $\mathcal{D}$ is the square of the $L_2$ distance. The samples closely resemble the original images, $\mathbf{x}_{\text{ori}}$s, from the MNIST testset, because each sample is positioned near an optimal point, and these optimal points are the original images, $\mathbf{x}_{\text{ori}}$s.

## 3.2 From Geometric Distance to Semantic Divergence

Based on the probabilistic perspective, we propose a semantic divergence, denoted by a non-symmetric divergence $\mathcal{D}_{\text{sem}}(\mathbf{x}_{\text{adv}}, \mathbf{x}_{\text{ori}}) := E(\mathbf{x}_{\text{adv}}; \mathbf{x}_{\text{ori}})$, where $E(\cdot; \mathbf{x}_{\text{ori}})$ represents the energy of an energy-based model trained on a dataset consisting of $\{t_1(\mathbf{x}_{\text{ori}}), t_2(\mathbf{x}_{\text{ori}}), \dots\}$. Here, $t_i \sim \mathcal{T}$, and $\mathcal{T}$ is a distribution of transformations that do not alter the original image's semantics. In practice, the choice of $\mathcal{T}$ depends on human subjectivity related to the dataset. Individuals are able to incorporate their personal comprehension of semantics into the model by designing their own $\mathcal{T}$. For instance, in the case of the MNIST dataset, the transformations could include scaling, rotation, distortion, and cropping, as illustrated in Figure 2(d). We assume that such transformations do not affect the semantics of the digits in the MNIST dataset. Consequently, our proposed semantic divergence induces the corresponding distance distribution $p_{\text{dis}}(\mathbf{x}_{\text{adv}}; \mathbf{x}_{\text{ori}}) \propto \exp(-c_1 \cdot E(\mathbf{x}_{\text{adv}}; \mathbf{x}_{\text{ori}}))$.

We claim that, given an appropriate $\mathcal{T}$, semantic divergence can surpass geometric distance. Empirically, maintaining the semantics of the original image by limiting the geometric distance between the adversarial image and the original image when deceiving a robust classifier is challenging: as shown in Figure 1 and Figure 3, it is difficult to preserve the semantics of the original images. The attacked images either display a 'shadow' of the target digits or reveal conspicuous tampering traces, such as in Figure 3(c), where the attacked digit turns gray. This phenomenon was empirically observed and tested by Song et al. [17] through an A/B test. Conversely, as depicted in Figure 4, the samples from $p_{\text{adv}}$ neither exhibit the 'shadow' of the target digits nor any obvious traces indicating adversarial attack. While semantic divergence can't entirely prevent the generation of a sample resembling the target class, as shown in Figure 4(a), we discuss certain techniques to mitigate this issue in Section 4.1.

A plausible explanation for this is that the utilization of geometric distance causes $p_{\text{dis}}(\cdot, \mathbf{x}_{\text{ori}})$ to overly focus on $\mathbf{x}_{\text{ori}}$. However, when applying semantic divergence induced by a suitable $\mathcal{T}$, the density of the distance distribution $p_{\text{dis}}(\cdot, \mathbf{x}_{\text{ori}})$ spreads out relatively more, resulting in a higher overlap between $p_{\text{dis}}(\cdot, \mathbf{x}_{\text{ori}})$ and $p_{\text{vic}}$. This, in turn, provides more opportunities for their product $p_{\text{adv}}$ to reach a higher value.

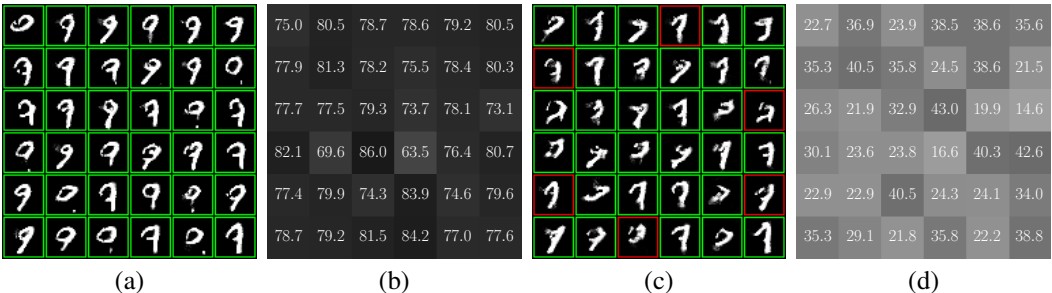

(a)            (b)            (c)            (d)

Figure 4: **(a) & (c)**: Samples from $p_{\text{adv}}(\cdot; \mathbf{x}_{\text{ori}}, y_{\text{tar}}) \propto \exp(-c_1 \cdot \mathcal{D}(\mathbf{x}_{\text{ori}}, \mathbf{x}_{\text{adv}})) \exp(-c_2 \cdot f(\mathbf{x}_{\text{adv}}, y_{\text{tar}}))$, where $\mathbf{x}_{\text{ori}}$ refers to the original image of digit "7" shown in Figure 1 and $y_{\text{tar}}$ refers to class 9. $\mathcal{D}$ represents our proposed semantic divergence. In (a), $f$ is the cross-entropy $f_{\text{CE}}$, while in (c), $f$ is $f_{\text{CW}}$. Constants are set as $c_1 = 1.0$ and $c_2 = 10^{-2}$. A green border indicates successful deception of the victim classifier, whereas a red border denotes failure. **(b) & (d)**: The predictive probability (softmax probability) of the target class, corresponding to each digit in Figures (a) and (c) on a one-to-one basis.

## 4   Deceiving robust classifiers

In this section, we present several techniques that enhance the performance of our proposed method in generating high-quality adversarial examples.

### 4.1   Victim distributions

The victim distribution $p_{\text{vic}} \propto \exp(c_2 \cdot f(\mathbf{x}_{\text{adv}}, y_{\text{tar}}))$ is influenced by the choice of function $f$. Let $g_\phi : [0,1]^n \to \mathbb{R}^{|\mathcal{Y}|}$ be a classifier that produces logits as output with $\phi$ representing the neural network parameters, $n$ denoting the dimensions of the input, and $\mathcal{Y}$ being the set of labels (the output of $g_\phi$ are logits). Szegedy et al. [18] suggested using cross-entropy as the function $f$, which can be expressed as

$$f_{\text{CE}}(\mathbf{x}, y_{\text{tar}}) := -g_\phi(\mathbf{x})[y_{\text{tar}}] + \log \sum_y \exp(g_\phi(\mathbf{x})[y]) = -\log \sigma(g_\phi(\mathbf{x}))[y_{\text{tar}}]$$

where $\sigma$ denotes the softmax function.

Carlini and Wagner [2] explored and compared multiple options for $f$. They found that, empirically, the most efficient choice of their proposed $f$s is:

$$f_{\text{CW}}(\mathbf{x}, y_{\text{tar}}) := \max(\max_{y \neq y_{\text{tar}}} g_\phi(\mathbf{x})[y] - g_\phi(\mathbf{x})[y_{\text{tar}}], 0).$$

From Figure 3 and Figure 4, we observe that $f_{\text{CW}}$ outperforms $f_{\text{CE}}$ when the $p_{\text{dis}}$ depends on either geometric distance or semantic divergence. A potential explanation for this phenomenon is that, according to its definition, $f_{\text{CW}}$ becomes 0 if the classifier is successfully deceived during the iteration process. This setting ensures that the generator does not strive for a relatively high softmax probability for the target class; it simply needs to reach a point where the victim classifier perceives the image as belonging to the target class. Consequently, after the iteration, the victim classifier assigns a relatively low predictive probability to the target class $\sigma(g_\phi(\mathbf{x}_{\text{adv}}))[y_{\text{tar}}]$, as demonstrated in Figure 3(d) and Figure 4(d).

In this study, we introduce two additional choices for the function $f$. Although these alternatives are not as effective as $f_{\text{CW}}$, we present them in Appendix C for further exploration.

### 4.2   Data Augmentation by Thin Plate Splines (TPS) Deformation

Thin-plate-spline (TPS) [1] is a commonly used image deforming method. Given a pair of control points and target points, TPS computes a smooth transformation that maps the control points to the target points, minimizing the bending energy of the transformation. This process results in localized deformations while preserving the overall structure of the image, making TPS a valuable tool for data augmentation.

As introduced in Section 3.2, we aim to train an energy-based model on transformations of a single image $\mathbf{x}_{\text{ori}}$. In practice, if the diversity of the augmentations of $\mathbf{x}_{\text{ori}}$, represented as $t(\mathbf{x}_{\text{ori}})$, is insufficient, the training of the probabilistic generative model is prone to overfitting. To address this issue, we use TPS as a data augmentation method to increase the diversity of $t(\mathbf{x}_{\text{ori}})$. For each $\mathbf{x}_{\text{ori}}$, we set a $5 \times 5$ grid of source control points, $\mathcal{P}_{\text{sou}} = \{(x^{(i)}, y^{(i)})\}_{i=1}^{5 \times 5}$, and defining the target points as $\mathcal{P}_{\text{tar}} = \{(x^{(i)} + \epsilon_x^{(i)}, y^{(i)} + \epsilon_y^{(i)})\}_{i=1}^{5 \times 5}$, where $\epsilon_x^{(i)}, \epsilon_y^{(i)} \sim \mathcal{N}(0, \sigma^2)$ are random noise added to the source control points. We then apply TPS transformation to $\mathbf{x}_{\text{ori}}$ with $\mathcal{P}_{\text{sou}}$ and $\mathcal{P}_{\text{tar}}$ as its parameters. This procedure is depicted in Figure 5. By setting an appropriate $\sigma$, we can substantially increase the diversity of the one-image dataset while maintaining its semantic content.

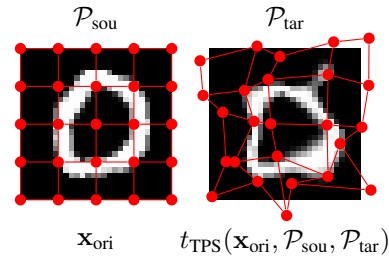

Figure 5: TPS as a data augmentation. **Left**: The original image $\mathbf{x}_{\text{ori}}$ superimposed with a $5 \times 5$ grid of source control points $\mathcal{P}_{\text{sou}}$. **Right**: The transformed image overlaid with a grid of target control points $\mathcal{P}_{\text{tar}}$.

### 4.3 Rejection Sampling

Directly sampling from $p_{\text{adv}}(\cdot; \mathbf{x}_{\text{ori}}, y_{\text{tar}})$ does not guarantee the generation of samples capable of effectively deceiving the classifier. To overcome this issue, we adopt rejection sampling [22], which eliminates unsuccessful samples and ultimately yields samples from $p_{\text{adv}}(\mathbf{x}_{\text{adv}} \mid \arg\max_y g_\phi(\mathbf{x}_{\text{adv}})[y] = y_{\text{tar}}; \mathbf{x}_{\text{ori}}, y_{\text{tar}})$.

### 4.4 Sample Refinement

After rejection sampling, the samples are confirmed to successfully deceive the classifier. However, not all of them possess high visual quality, as demonstrated in Figure 4(c). To automatically obtain $N$ semantically valid samples[1], we first generate $M$ samples from the adversarial distribution. Following rejection sampling, we sort the remaining samples and select the top $\kappa$ percent based on the softmax probability of the original image's class, as determined by an auxiliary classifier. Finally, we choose the top $N$ samples with the lowest energy $E$, meaning they have the highest likelihood according to the energy-based model.

The auxiliary classifier is trained on the data-augmented training set. We do not use the energy of the samples as the sole criterion for selection because some low-visual quality samples may also have a high likelihood. This occurrence is further explained and examined in Appendix D. The entire process of rejection sampling and sample refinement is portrayed in Algorithm 1.

---

**Algorithm 1** Rejection Sampling and Sample Refinement

---

**Input:** A trained energy based model $E(\cdot; \mathbf{x}_{\text{ori}})$ based on the original image $\mathbf{x}_{\text{ori}}$, the victim classifier $g_\phi$, an auxiliary classifier $g_\psi$, number of initial samples $M$, number of final samples $N$, the percentage $\kappa$.
**Output:** $N$ adversarial samples $\mathbf{x}$.
    $\mathbf{x} = \emptyset$
    **for** $0 \leq i < M$ **do**
        $\mathbf{x}_{\text{adv}} \sim p_{\text{adv}}(\cdot; \mathbf{x}_{\text{ori}}, y_{\text{tar}})$                 ▷ Sample from the adversarial distribution.
        **if** $\arg\max_y g_\phi(\mathbf{x}_{\text{adv}})[y] = y_{\text{tar}}$ **then**          ▷ Accept if $\mathbf{x}_{\text{adv}}$ deceive the classifier.
            $\mathbf{x} = \mathbf{x} \cup \{\mathbf{x}_{\text{adv}}\}$
        **end if**
    **end for**
    Sort $\mathbf{x}$ by $\sigma(g_\psi(\mathbf{x}_i))[y_{\text{ori}}]$ for $i \in \{1, \ldots, |\mathbf{x}|\}$ in descent order
    $\mathbf{x} = (\mathbf{x}_i)_{i=1}^{\lfloor \kappa |\mathbf{x}| \rfloor}$                               ▷ Select the first $\kappa$ percent elements from $\mathbf{x}$.
    Sort $\mathbf{x}$ by $E(\mathbf{x}_i; \mathbf{x}_{\text{ori}})$ for $i \in \{1, \ldots, |\mathbf{x}|\}$ in ascent order
    $\mathbf{x} = (\mathbf{x}_i)_{i=1}^{N}$                                   ▷ Select the first $N$ elements from $\mathbf{x}$.

---

---

[1]In practice, we could select adversarial samples by hand, but we focus on automatic selection here.

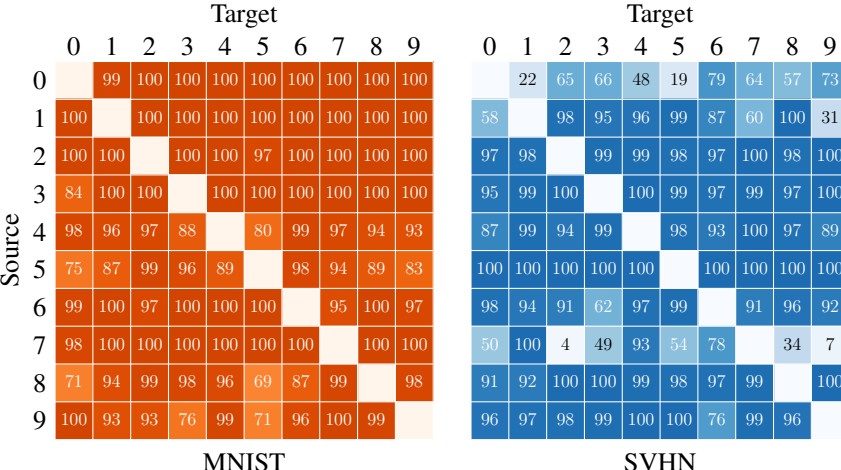

Figure 6: The success rates (%) of our targeted unrestricted adversarial attack. Corresponding sample examples for each grid are depicted in the top right and bottom right sections of Figure 1. Refer to Table 1 for overall success rate.

## 5 Experiment

### 5.1 Implementation

We implemented our proposed semantics-aware adversarial attack on two datasets: MNIST and SVHN. For the MNIST dataset, the victim classifier we used was an adversarially trained MadryNet [12]. For the SVHN dataset, we utilized an adversarially trained ResNet18, in accordance with the methodology outlined by Song et al. [17]. On the distance distribution side, for every original image denoted as $\mathbf{x}_{\text{ori}}$, we trained an energy-based model on the training set, which is represented as $\{t_1(\mathbf{x}_{\text{ori}}), t_2(\mathbf{x}_{\text{ori}}), \dots\}$. In this case, $t_i$ follows a distribution of transformations, $\mathcal{T}$, that do not change the semantics of $\mathbf{x}_{\text{ori}}$. For the MNIST dataset, we characterized $\mathcal{T}_{\text{MNIST}}$ as including Thin Plate Spline (TPS) transformations, scaling, and rotation. For the SVHN dataset, we defined $\mathcal{T}_{\text{SVHN}}$ as comprising Thin Plate Spline (TPS) transformations and alterations in brightness and hue. Detailed specifics related to our implementation can be found in Appendix A.

### 5.2 Evaluation

Our method generates adversarial samples that can deceive classifiers, but it does not guarantee the preservation of the original label's semantic meaning. As such, we consider an adversarial example successful if human annotators perceive it as having the same meaning as the original label, in line with the approach by Song et al. [17]. To enhance the signal-to-noise ratio, we assign the same image to five different annotators and use the majority vote as the human decision, as done in [17]. The screenshot of the annotator's interface is in Appendix B.

In detail, we begin with an original image $\mathbf{x}_{\text{ori}}$, its label $y_{\text{ori}}$, and a target class $y_{\text{tar}}$. We draw $M = 2000$ samples from $p_{\text{adv}}(\cdot; \mathbf{x}_{\text{ori}}, y_{\text{tar}})$, rejecting those that fail to deceive the victim classifier. After sample refinement, we obtain $N = 100$ adversarial examples, $\mathbf{x}_{\text{adv}}^{(i)}$ for $i \in \{1, \dots, N\}$. We express the human annotators' decision as function $h$ and derive the human decision $y_{\text{hum}}^{(i)} = h(\mathbf{x}_{\text{adv}}^{(i)})$. As previously mentioned, an adversarial example $\mathbf{x}_{\text{adv}}^{(i)}$ is considered successful if $y_{\text{hum}}^{(i)}$ is equal to $y_{\text{ori}}$. We then compute the success rate $s$ as follows:

$$s = \frac{\sum_{i=1}^{N} \mathbb{1}(y_{\text{hum}}^{(i)} = y_{\text{ori}})}{N}$$

where $\mathbb{1}$ represents the indicator function.

We randomly select 10 digits, each representing a different class, from the MNIST/SVHN test set to serve as the original image $\mathbf{x}_{\text{ori}}$. These are depicted on the left side of Figure 1. For each $\mathbf{x}_{\text{ori}}$, we iterate through the target class $y_{\text{tar}}$ ranging from 0 to 9, excluding the class $y_{\text{ori}}$ that signifies

Table 1: Success rate comparison between the method proposed by Song et al. [17] and ours. The results presented in this table are for reference only, as Song's results are taken directly from their paper, and we did not use the same group of annotators for our evaluation.

| Robust Classifier | Success Rate of Song et al. [17] | Our Success Rate |
|---|---|---|
| MadryNet [12] on MNIST | 85.2 | **96.2** |
| ResNet18 [8] (adv-trained) on SVHN | 84.2 | **86.3** |

the ground-truth label of $\mathbf{x}_{\text{ori}}$. As previously described, for every pair of $\mathbf{x}_{\text{ori}}$ and $y_{\text{tar}}$, we generate $N = 100$ adversarial examples post sample refinement. The result of each pair is illustrated in Figure 6. The overall success rate is illustrated in Figure 1.

## 5.3 Results

As depicted in Figure 6 and Table 1, our proposed method often succeeds in fooling robust classifiers, all the while preserving the original semantics of the input. It should be noted, however, that this does not occur in every instance.

## 6 Related work

**Unrestricted adversarial examples**  Song et al. [17] proposed generating unrestricted adversarial examples from scratch using conditional generative models. In their work, the term "unrestricted" indicates that the generated adversarial samples, $\mathbf{x}_{\text{adv}}$, are not restricted by a geometric distance such as the $L_2$ norm or $L_\infty$ norm. The key difference between their approach and ours is that their adversarial examples $\mathbf{x}_{\text{adv}}$ are independent of any specific $\mathbf{x}_{\text{ori}}$, while our model generates $\mathbf{x}_{\text{adv}}$ based on a given $\mathbf{x}_{\text{ori}}$. By slightly modifying (7), we can easily incorporate Song's "unrestricted adversarial examples" into our probabilistic perspective:

$$p_{\text{adv}}(\mathbf{x}_{\text{adv}}; y_{\text{sou}}, y_{\text{tar}}) := p_{\text{vic}}(\mathbf{x}_{\text{adv}}; y_{\text{tar}}) p_{\text{dis}}(\mathbf{x}_{\text{adv}}; y_{\text{sou}}) \tag{8}$$

where $y_{\text{sou}}$ is the source class. It becomes evident that the adversarial examples generated by our $p_{\text{adv}}(\cdot; \mathbf{x}_{\text{ori}}, y_{\text{tar}})$ adhere to Song's definition when $\mathbf{x}_{\text{ori}}$ is labeled as $y_{\text{sou}}$.

**TPS as a Data Augmentation Technique**  To the best of our knowledge, Vinker et al. [21] were the first to employ TPS as a data augmentation method. They utilized TPS as a data augmentation strategy in their generative model for conditional image manipulation based on a single image.

## 7 Limitation

This work's foremost limitation pertains to the inherent difficulties in training energy-based models (EBMs), as underscored in the earlier studies by Du and Mordatch [4] and Grathwohl et al. [7]. The EBM training process is notoriously challenging, and a notable gap persists between the generation quality of EBMs and that of other widely-used probabilistic generative models, such as variational autoencoders and diffusion models. Consequently, we are currently unable to generate adversarial samples for images with higher resolution.

## 8 Conclusion

In this work, we present a probabilistic perspective on adversarial examples by employing Langevin Monte Carlo. Building on this probabilistic perspective, we introduce semantic divergence as an alternative to the commonly used geometric distance. We also propose corresponding techniques for generating semantically-aware adversarial examples. Human participation experiments indicate that our proposed method can often deceive robust classifiers while maintaining the original semantics of the input, although not in all cases.

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
