# OpenReview forum: "Constructing Semantics-Aware Adversarial Examples with Probabilistic Perspective"
_NeurIPS.cc/2023/Conference — Submitted to NeurIPS 2023_

### Official Review · Reviewer_AGWj · 2023-06-11

**Soundness:** 3 good
**Presentation:** 3 good
**Contribution:** 3 good
**Rating:** 6
**Confidence:** 4

**Summary:**


This paper introduces a novel approach to adversarial attacks that goes beyond traditional norm bounded attacks. Instead, the proposed method focuses on unrestricted attacks that are both effective and capable of preserving the semantic meaning of the input data.

The method utilizes Langevin Monte Carlo techniques to sample from a distribution of potential attacks. To ensure semantic preservation, a learned energy function is employed, which guides the generation of adversarial samples. Rejection sampling and refinement techniques are then applied to select and further improve the quality of the generated samples.

The evaluation of the proposed method demonstrates a significant success rate when attacking defended models. By allowing for unrestricted attacks while maintaining semantic integrity, this approach presents a promising advancement in the field of adversarial attacks, showcasing its effectiveness and potential for practical application.

**Strengths:**

1. Interesting work on unrestricted adversarial attack, which is important given that most attacks now are bounded attack.

2. The method is effective in breaking already defended models. Fig 1,2 clearly shows the advantage over norm bounded attacks.

**Weaknesses:**

1. What is the computation cost of the attack? The paper only evaluates on two toy datasets, MNIST and SVHN, the reviewer is wondering if the method can generalize to larger dataset.

2. Ablation study on the component is missing. Like TPS as data augmentaion, the effect of the choice of the sampling method. Also the method requires specify several hyper parameters, like M. Ablation study is useful.

**Questions:**

Is the TPS augmentation used to capture the energy function of the semantic?

Would TPS augmentation also work for other type of data, say semantic segmentation, where the location of the pixel matters a lot?

Similar constraint functions are used for defending adversarial attacks, such as [1,2], but they use the constraint for defense. Can the author discuss if their attack can attack the dynamic defense in [1,2], where the defense will reverse the attack to the benign manifold?

[1] Mao et al. Adversarial Attacks are Reversible with Natural Supervision. ICCV 2021.

[2] Mao et al. Robust Perception through Equivariance. ICML 2023.

---

> ### Author Rebuttal · Authors · 2023-08-10
>
> Thanks for reviewing! Below is our response:
>
> ### Weaknesses
> 1. This attack costs more than traditional methods because we have to fit an energy-based model for each instance if we want to generate adversarial example based on this instance. In our global response, we added CIFAR10 experiment.
>
> 2. For the TPS augmentation, the noise follows a Gaussian distribution with a variance of 0.01. We refrained from conducting an ablation study on this parameter as an inappropriate selection could disrupt the training of the energy-based model. As for M, due to resource constraints, particularly the need for annotators to label the data, we were unable to perform an ablation study.
>
> ### Questions
> 1. The primary advantage of TPS augmentation is that it augments the dataset with similar data, facilitating a smoother training process for the EBM.
>
> 2. Yes, exactly. TPS augmentation is a commonly employed technique in image segmentation tasks.
>
> 3. Thank you for introducing these studies to me! There's a profound intrinsic connection between these works and ours. While they focus on the data manifold, we emphasize the data distribution. This distinction results in them utilizing the contrastive loss as a constraint function, while we employ probability. We are confident that our attack could effectively challenge this dynamic defense because **our generated adversarial examples maintain the essence of what they are attempting to reverse**.
>
>
> #### Reference
>
> [1] Mao et al. Adversarial Attacks are Reversible with Natural Supervision. ICCV 2021.
>
> [2] Mao et al. Robust Perception through Equivariance. ICML 2023.

---

> > ### Comment · Reviewer_AGWj · 2023-08-19
> > **Thank for the rebuttal.**
> >
> > The reviewer thanks the authors for the rebuttal.
> >
> > While more interesting studies can be done in the future to address the limitations, the reviewer thinks this is a good initial step to introduce this new type of attack.

---

> ### Author Response · Authors · 2023-08-18
> **Better Visual Result on CIFAR-10**
>
> Echoing the suggestions from reviewers oacq and tkih, we recognized the need to enhance the visual results on CIFAR-10. Through further experimentation, we found that by reducing the perturbation magnitude of TPS and incorporating scaling into $\mathcal{T}$, we achieved more visually appealing results for CIFAR-10. Adhering to the submission guidelines, I can't provide direct images or links here. Nonetheless, I've submitted the improved visuals to the area chair for review, and I anticipate that you will be able to access them soon. We hope these updates address your concerns more comprehensively.
>
> Your insights regarding TPS adjustment have been invaluable during this phase, and we are truly appreciative of your guidance. Nonetheless, it's worth noting again that a comprehensive ablation study on TPS's parameters was not feasible for us. Reducing the perturbation of the data makes the energy-based model more challenging to train, as highlighted in [1, 2].
>
> #### Reference
>
> [1] Song, Yang, and Diederik P. Kingma. "How to train your energy-based models." arXiv preprint arXiv:2101.03288 (2021).
>
> [2] Grathwohl, Will, et al. "Your classifier is secretly an energy based model and you should treat it like one." arXiv preprint arXiv:1912.03263 (2019).

---

### Official Review · Reviewer_C5cG · 2023-06-30

**Soundness:** 3 good
**Presentation:** 2 fair
**Contribution:** 3 good
**Rating:** 5
**Confidence:** 4

**Summary:**

The adversarial examples generated by classical methods such as PGD have different semantic meaning to the original label, which means that the adversarial examples are easy to be distinguished by human. In this paper, the authors focus on the generalization of adversarial example which preserves the original semantic information. They propose a semantically-aware distance measure to replace the geometrical distance measure. And they use Langevin Monte Carlo method to find the minimal point (adversarial sample) of their proposed loss function. Several techniques that further enhance the performance of the proposed method are presented. From the experimental results, it seems that their generated examples preserve the original semantical imformation.

**Strengths:**

* As far as I know, the proposed adversarial attack method is novel.
* They proposed a semantical distance measure to generate the semantic-aware examples. Although the idea of semantical measure already exists in many previous work, I think the usage here in adversarial example generalization scenario is interesting and reasonable.
* Their method is theoretically and experimentally reliable.

**Weaknesses:**

* One of the limitation of this paper is that, the loss of semantics of adversarial examples only exists in some simple tasks, such as MNIST and SVHN. As the experimental results in previous work shows, the adversarial examples of CIFAR and ImageNet have very little disturbations that cannot be distinguished by human and preserve the semantical information. Hence, I think the significance of this paper is somewhat limited.
* The motivation of using EBMs and LMC is not very clear to me. In my opinion, we can directly optimize the semantic-aware loss to generate the adversarial examples. The necessity of using the EBMs and LMC should be stated more clearly.
* In the experiment part, the success rate involves subjective factors. They use human annotators to determine whether the adversarial examples have the same meaning as the original label. Is there a more subjective metric? Otherwise, the experimental results may suffer a credibility crisis.
* More experiments on CIFAR-10 and CIFAR-100 are necessary.
* Can you give a more detailed explaination of the training of the energy-based model? I noticed that Section 2.5 includes some brief introduction, but what is the data distribution $p_d$ here? What is the specific training algorithm?

If the authors can address my concerns well, I will consider raise the score.

**Questions:**

* I am confused when I read Line 69-70. Should $exp(g(x))$ be replaced by $exp(-g(x))$? Otherwise, the distribution $p(x)$ seems to concentrate around the global maximal.
* If we use the semantric-aware adversarial examples to adversarially train the model, will the model be robust to the semantric-aware adversarial example?


**Limitations:**

Yes.

---

> ### Author Rebuttal · Authors · 2023-08-10
>
> Thanks! Here are our responses to each of the points you've raised in your concerns:
>
> ### Weaknesses
> - Referring to Figure 1 in our global response PDF, the adversarial samples produced by PGD on CIFAR retain their semantics. However, the attacked images exhibit visual features, such as unnatural colors, which make them potentially detectable by humans.
> - Indeed, one can directly optimize the loss which is aware of semantics to produce adversarial examples. Yet, in this research, we introduce a principle probabilistic model from which adversarial examples can be sampled. The unnormalized adversarial distribution we've proposed employs LMC as an intuitive method for sampling from this distribution. Subsequently, the EBM is integrated to replace the distance distribution $p_{dis}$, thereby implicitly representing a semantic distance. We opt for EBM over other probabilistic models primarily because EBM inherently aligns well with LMC. For a more in-depth discussion, please refer to Question 2 by reviewer oacq.
> - In addition, we assess the transferability of our proposed method in our global response.
> - We've incorporated a CIFAR10 experiment in our global response.
> - We use the same training method as Du et al's work [1]. If we want to fit $p_{dis}(x_{adv}; x_{ori})$ by an energy-based model, then the dataset for training the energy based model is $\\{ t_1(x_{ori}), t_2(x_{ori}), ... \\}$, a dataset generated by a single image $x_{ori}$. The training details are introduced in [1].
>
> ### Questions
> - Thank you for pointing out the typo. The term $-E_\theta$ in line 70 should be correctly noted as $E_\theta$.
> - In our scenario, the outcome hinges on the choice of $\mathcal{T}$ as introduced in line 122. If a model's training phase includes adversarial examples induced by $\mathcal{T}$ , then this model will exhibit robustness to adversarial examples generated by the same set of transformations $\mathcal{T}$. However, it's unlikely that the model will remain robust against adversarial attacks prompted by a different set of transformations, $\mathcal{T}'$.
>
> #### Reference
> [1] Du, Yilun, and Igor Mordatch. "Implicit generation and generalization in energy-based models." arXiv preprint arXiv:1903.08689 (2019).

---

> > ### Comment · Reviewer_C5cG · 2023-08-16
> >
> > Thank the authors for their responses. Their response partially addressed my concerns and I maintain the rating.

---

> > > ### Author Response · Authors · 2023-08-17
> > > **Better Visual Result on CIFAR-10**
> > >
> > > Thank you for your feedback! Echoing the suggestions from reviewers oacq and tkih, we recognized the need to enhance the visual results on CIFAR-10. Through further experimentation, we found that by reducing the perturbation magnitude of TPS and incorporating scaling into $\mathcal{T}$, we achieved more visually appealing results for CIFAR-10. Adhering to the submission guidelines, I can't provide direct images or links here. Nonetheless, I've submitted the improved visuals to the area chair for review, and I anticipate that you will be able to access them soon. We hope these updates address your concerns more comprehensively.

---

### Official Review · Reviewer_tkih · 2023-07-08

**Soundness:** 3 good
**Presentation:** 3 good
**Contribution:** 3 good
**Rating:** 5
**Confidence:** 5

**Summary:**

This paper proposes to generate semantics-preserving adversarial examples by framing the construction of adversarial examples as a box-constrained non-convex optimization problem. More specifically, the authors propose a Langevin Monte Carlo (LMC) technique to craft adversarial examples that preserve the meaning of the original inputs they are derived from. With this framing, they cast the generation of adversarial examples as a semantic-based probabilistic distribution. The authors showed that their semantic-aware adversarial attack is capable of fooling robust classifiers while preserving most of the semantics of their source images.

**Strengths:**

This paper is quite interesting paper and well-written. The problem is well-defined, and the solution quite intuitive. The math is also quite sound. Although the problem of generating semantics-preserving adversarial examples has been studied extensively in the past, it still remains relevant. This paper proposes another interesting perspective on how to approach this problem.

**Weaknesses:**

Although the paper is interesting, the evaluation is quite limited. For instance, the approach is only evaluated on MNIST and SVHN. Evaluating the approach against "more challenging" datasets like ImageNet, CIFAR-10, CIFAR-100 would make their contributions more compelling. Also, studying the transferability property of their attacks would strengthen their paper, and give more confidence to the readers about the strength of their attacks. Moreover, I would have liked to see how the magnitude of the noise used in Thin-plate-spine affects the overall performance of their attacks. Finally, the related work section is rather limited. There is a plethora of interesting studies in crafting adversarial examples that are semantics-preserving. For instance, [1] and [2] are quite related to the approach the authors propose, and should be evaluated or discussed further in the related work section.

[1]: Semantics Preserving Adversarial Examples. https://aisecure-workshop.github.io/amlcvpr2021/cr/27.pdf
[2]: Localized Uncertainty Attacks. https://ui.adsabs.harvard.edu/abs/2021arXiv210609222A/abstract


**Questions:**

I would highly recommend the authors to further experiment with datasets like ImageNet, CIFAR-10, CIFAR-100, etc., to study the transferability property of their adversarial attacks, and to improve the related work section by comparing their approach against relevant approaches that were proposed in the past.

**Limitations:**

Yes.

---

> ### Author Rebuttal · Authors · 2023-08-10
>
> Thanks for reviewing! Below is our response:
>
> > ... the approach is only evaluated on MNIST and SVHN ... Also, studying the transferability property of their attacks would strengthen their paper, and give more confidence to the readers about the strength of their attacks.
>
> We've incorporated a CIFAR10 experiment and evaluated transferability in our global response.
>
> >  Moreover, I would have liked to see how the magnitude of the noise used in Thin-plate-spine affects the overall performance of their attacks.
>
> The noise is Gaussian distribution with variance 0.01. We do not do any ablation study on this parameter because the a bad choice of this parameter may clash the training of energy based model.
>
> >  Finally, the related work section is rather limited.
>
> In light of your feedback and input from other reviewers, we will provide a more comprehensive discussion on related work in our updated version.

---

> > ### Comment · Reviewer_tkih · 2023-08-14
> >
> > I thank the authors for the detailed rebuttal and appreciate the additional experiments they ran. After carefully examining the adversarial examples generated from the CIFAR-10 dataset, it's fair to say that this method is not that semantics-preserving as the images appear quite distorted. The paper is still interesting nonetheless. Maybe some of the claims could be watered down a bit, and the limitations clearly specified in the manuscript.

---

> > > ### Author Response · Authors · 2023-08-14
> > > **Feedback Response**
> > >
> > > Thank you for your feedback. Based on our understanding, Figure 1 in the attached PDF suggests that **TPS might not be the optimal data augmentation method for preserving CIFAR10's semantics**. However, this does not invalidate our overall approach.
> > >
> > > As mentioned in section 3.2 of the submitted paper:
> > > > In practice, the choice of $\mathcal{T}$ depends on human subjectivity related to the dataset. Individuals are able to incorporate their personal comprehension of semantics into the model by designing their own $\mathcal{T}$.
> > >
> > > If we consider TPS as a suitable method for preserving CIFAR10's semantics, meaning that the distortions introduced by TPS don't hinder our understanding of the image's intent, then Figure 1 in our attached PDF is in alignment with our intentions.
> > >
> > > While distortions in hand-written digits don't inhibit our ability to identify the digit, the images on the right side of Figure 1 might appear unnatural to human viewers. This is especially true for objects that typically have a defined structure, such as cars, trucks, and ships. Additionally, the perspective from which the object is viewed can influence this perception.
> > >
> > > Our assertion that we "transcends the restriction imposed by geometric distance, instead opting for semantic constraints" is underpinned by the mathematical framework presented in section 3.1. The perceptual incongruities evident in Figure 1 arise primarily from the choice of TPS as a data augmentation method and its potential effects on semantics, rather than an inherent flaw in our proposed method.
> > >
> > > Our method provides a pathway for individuals to embed their subjective understanding of semantics via data augmentation, represented by $\mathcal{T}$. Yet, if this interpretation doesn't resonate with general human semantic perception, the resulting images may be suboptimal. As you've pointed out, we will highlight this nuance in the limitations section of our paper.

---

> > > > ### Comment · Reviewer_tkih · 2023-08-16
> > > >
> > > > There seems to be a "consensus" in the research community that semantics preservation means the adversarial examples one crafts must look natural and visually similar to their clean counterparts. I worked on a similar problem in the past. It is always difficult to convince reviewers of the soundness of one's method if the adversarial examples present (serious) distortions that make them appear unnatural. Here, the distortions are quite noticeable. I am raising my rating from Borderline Reject to Borderline Accept, however, because the idea is interesting, but the soundness of the adversarial examples remains an issue. I hope the chairs will also weigh in and decide on the outcome of this submission in light of its strength and weaknesses.

---

> > > > > ### Author Response · Authors · 2023-08-17
> > > > > **Better Visual Result on CIFAR-10**
> > > > >
> > > > > Thank you for recognizing the merit in our idea, and for your continued support and encouragement.
> > > > >
> > > > > We concur with the consensus that preserving semantics implies that generated adversarial examples should closely resemble their original, unaltered counterparts in appearance. To this end, we conducted further experiments. By reducing the intensity of the TPS perturbation and incorporating scaling into our data augmentation $\mathcal{T}$, we've achieved notably enhanced visual results for CIFAR-10. We believe these refined results mark a considerable advancement from our earlier version. In line with the guidelines, I'm unable to directly include any images or links in this message. However, I've forwarded the updated visuals to the area chair, and I expect you'll be able to review them shortly.

---

### Official Review · Reviewer_oacq · 2023-07-16

**Soundness:** 1 poor
**Presentation:** 3 good
**Contribution:** 2 fair
**Rating:** 5
**Confidence:** 4

**Summary:**

In this work, a probabilistic view of adversarial examples based on the [projected stochastic gradient Langevin algorithm](https://proceedings.mlr.press/v134/lamperski21a.html) is introduced and used as an optimization algorithm instead of the SGD or Adam optimizer for adversarial examples. In addition, the geometric constraint (Lp norms) is replaced by a semantic distance criterion based on an instance-wise energy-based model (i.e., an EBM is trained for each instance, using transformed versions as the training dataset) to ensure semantic/visual proximity to the original input. They improved the adversarial examples using the [CW objective](https://www.computer.org/csdl/proceedings-article/sp/2017/07958570/12OmNviHK8t) and thin-plate splines transformation to create a more diverse training dataset for EBM training. Moreover, they generated a set of successful adversarial attacks (i.e., fooled the classifier) via rejection sampling and proposed a simple selection procedure to select the final adversarial examples based on the softmax probabilities of an auxiliary classifier and the energy of the examples. The experiments show that the proposed method is able to generate adversarial examples that fool the classifier while being visually/semantically indistinguishable to humans.

**Strengths:**

- The proposed method is very detailed and intricate.
- The Langevin Monte Carlo-based optimization procedure seems to improve the quality of adversarial examples overall.
- The paper is well-written and clearly structured.
- Code is provided.

**Weaknesses:**

- Previous work, e.g., by [Sharma & Chen](https://openreview.net/forum?id=Sy8WeUJPf), has also generated visually similar adversarial examples for the MadryNet while still using a geometric distance ([elastic-net regularization](https://arxiv.org/abs/1709.04114)). This raises questions about the generality of the work’s central claim that it “transcends the restriction imposed by geometric distance, instead opting for semantic constraints” (L4-5) beyond the limitations of the adversarial attack methods shown in the present work.
- The present work only shows experiments on digit-based datasets (MNIST & SVHN). Applications to datasets with natural images (e.g., CIFAR or ImageNet) are missing. Consequently, the necessity and applicability of the proposed adversarial attack are very unclear, since for natural images the adversarial examples typically remain visually very close to the original inputs; also after adversarial fine-tuning.
- The work is missing interesting experiments, e.g., what would happen if we use the proposed adversarial attack approach for adversarial training? Does it improve adversarial robustness? Does the adversarial attack also bypass certified defenses? Overall, the experimental section is very short (3 lines of results) and would greatly benefit from, e.g., the aforementioned experiments.
- The approach requires an instance-wise energy-based model for its semantic distance loss, which must be trained for every sample (on different augmented versions); cf. L122. This may limit its applicability.
- The proposed attack and problem setup are not quite original, i.e., it combines well-known techniques, or previous work (see first point above) has also already targeted the visual similarity challenge of adversarial examples for adversarially fine-tuned models.

**Questions:**

- Couldn't we just train a single energy-based model for the specific data domain? If so, how do the generated adversarial examples compare to those using an instance-based energy-based model for semantic divergence?
- Why do the authors refrain from using (currently) better generative methods?
- Regarding Fig. 2: are the samples generated using only the classifier (neglecting the distance distribution term) for a & b and vice versa for c?

## Suggestions

- The related work could include more discussion on previous works on adversarial examples.
- For the minimization problem formulations in Sec. 2.1., it’d be good to include that $x_{adv}$ is minimized (even though it’s obvious given the work’s scope).
- It’d be meaningful to include error bars for the experiments.
- Results for Song et al in Tab. 1 should be repeated for better comparability, if possible.

**Limitations:**

The limitations are adequately addressed.

---

> ### Author Rebuttal · Authors · 2023-08-08
>
> Thanks! Here are our responses to each of the points you've raised in your concerns:
>
> ### Weakness
>
> - The assertion that our method 'transcends the restriction imposed by geometric distance' is based on a theoretical perspective, as outlined in lines 112-116. In this context, geometric distances lead to certain specific distributions for $p_{dis}$. For instance, an L2 norm corresponds to a Gaussian distribution, while an L1 norm aligns with a Laplace distribution. The elastic-net attack you mentioned leverages a weighted sum of L1 and L2 distances, which in turn leads to a weighted product of a Laplace distribution and a Gaussian distribution for $p_{dis}$. Indeed, by selecting an optimal weight $\beta$, the elastic-net strategy demonstrates effective results in attacking Madrynet, thereby highlighting the good performance of the induced $p_{dis}$. However, the $p_{dis}$ that arises from geometric distance represents just a minuscule portion of all possible $p_{dis}$ choices. We posit that as probabilistic generative models continue to evolve, data-driven $p_{dis}$ models should be able to provide superior performance.
> - We have incorporated CIFAR10 experiments in our global response, see Figure 1 of the attached PDF.
> - Kindly consult Table 1 and Table 2 in the PDF attached to our global response. Our technique can successfully circumvent certified defenses. Additionally, we showcase the transferability of our proposed attack strategy in these tables. We refrained from incorporating our generated adversarial examples into a subsequent adversarial training process. This is because our attack relies on the data augmentation, denoted as $\mathcal{T}$. Consequently, we don't anticipate that a newly adversarially trained model would exhibit enhanced defense against conventional attacks.
> - Indeed, training an energy-based model for each $x_{ori}$ is a limitation of our study. However, we believe there are scenarios where generating a few adversarial examples is both sufficient and crucial. Moreover, we have also proposed a method requiring only one energy-based model per domain, as highlighted in the first bullet point of the 'Questions' section of this response.
> - We firmly believe that our proposed methodology stands out both in principle and novelty. Our approach introduces a fresh, elegant probabilistic perspective by factoring the adversarial distribution into $p_{vic}$ and $p_{dis}$. The notion of $p_{vic}$ resonates with the idea that "samples can be drawn from adversarially trained classifiers", while $p_{dis}$ aligns with the concept of geometric distance, especially when pertaining to Gaussian or Laplace distributions. Moreover, the technique of using a probabilistic model to fit $p_{dis}$ is also a groundbreaking addition. While numerous studies employ generative models in the adversarial domain, we assert that our method offers a distinctive blend of generative modeling and adversarial attack, making it both innovative and elegant.
>
> ### Questions
> - Yes, we can.  If we consider each domain as a class, the setup aligns perfectly with Song's configuration, as depicted in formula (8). For a visual representation of the unrestricted adversarial examples generated under this setting, please see Figure 2a in our global response PDF.
> - We use EBM because it can directly model the unnormalized distribution (formula (3)). For other popular generative models, GANs can not provide $p(x)$, VAEs can only give a lower bound of $p(x)$, and diffusion models attempt to fit a probabilistic model based on a noise-altered $x$, denoted as $p(\tilde{x})$. While both Normalizing flow and PixelCNN can provide $p(x)$, practically speaking, generating quality samples through Langevin dynamics on their gradient $\nabla\log p(x)$ is challenging. Although these challenges might be mitigated with certain modifications, our paper's primary objective is to introduce a novel probabilistic perspective on adversarial attacks. Utilizing Langevin dynamics offers a straightforward method for sampling from $p_{adv}$, and employing the EBM to model $p_{dis}$ ensures precision and elegance throughout the model. While this might not guarantee peak performance, we believe it establishes a strong foundation for this model series.
> - Yes, exactly. Adversarially trained classifiers have generation ability; cf. L110. That is why decomposing $p_{adv}$ into $p_{vic}$ and $p_{dis}$ is a logical approach.
>
> ### Suggestions
> - We'll expand our discussion on related work in response to your feedback and suggestions from other reviewers.
> - This will be included in our updated version.
> - In line with the evaluation methodology of Song et al.'s work, we consolidated the results of five annotators. Therefore, providing error bars at this juncture isn't feasible for us.
> - Based on their GitHub repository, it appears that Song et al.'s work cannot be replicated precisely due to the absence of the adversarial training component.

---

> > ### Comment · Reviewer_oacq · 2023-08-14
> > **Response to rebuttal**
> >
> > I thank the authors for their detailed rebuttal. Specifically, I appreciate the additional CIFAR-10 experiments, the extension of EBMs from an instance-wise to a class-wise attack design and bypassing geometric certified defenses. Below, I address some of my concerns that still persist after the authors' rebuttal.
> >
> > > Severely deformed visual results for CIFAR-10 (e.g., car or horse class)
> >
> > Although I appreciate the experiments on CIFAR-10, they unfortunately provide empirical confirmation of one of my main concerns that the method does not work as well on natural images and thus violates the main goal of this work (“our semantics-aware adversarial attack is capable of deceiving robust classifiers while preserving most of the original image’s semantics” L32-33). While I tend to agree with the former (“deceiving”), I disagree with the latter (“preserving”). For example, the CIFAR classes for cats, dogs, horses, ships, and trucks are severely deformed (interestingly with a similar blurred curve-like pattern?) but PGD with L2 norm is also able to reliably deceive the classifier (except for some truck examples). In my personal evaluation, I would have chosen PGD with L2 norm as the most semantically preserving in Figure 1 in at least 6 out of 10 cases. I am very confident that other people would have similar choices.
> >
> > > “Moreover, the technique of using a probabilistic model to fit [a distance distribution] is also a groundbreaking addition”
> >
> > I tend to disagree and kindly refer to some previous works, e.g., [1]. However, I acknowledge that the practical application of a semantic distance loss to adversarial examples is original; as already mentioned in my original review.
> >
> > > “The assertion that our method 'transcends the restriction imposed by geometric distance' is based on a theoretical perspective, as outlined in lines 112-116.”
> >
> >  I was/am aware of these mentioned lines but there are many places of the present manuscript, where it sounded more like a fundamental limitation of Lp distance norms, e.g., L18-22 (introduction). But it is not, as also acknowledged by the authors. Thus, I’d suggest clarifying such statements because it may mislead readers of the work.
> >
> > ---
> >
> > [1] Larsen, Anders Boesen Lindbo, et al. "Autoencoding beyond pixels using a learned similarity metric." ICML 2016.

---

> > > ### Author Response · Authors · 2023-08-15
> > > **Addressing concerns**
> > >
> > > Thank you for your feedback! Please allow us to provide further clarification:
> > >
> > > ### The CIFAR10 result
> > >
> > > > In my personal evaluation, I would have chosen PGD with L2 norm as the most semantically preserving in Figure 1 in at least 6 out of 10 cases. I am very confident that other people would have similar choices.
> > >
> > > We acknowledge that the images presented on the right side of Figure 1 might not resonate with human semantic perceptions. However, this doesn't invalidate the methodology we proposed.
> > >
> > > As mentioned in section 3.2 of the submitted paper:
> > > > In practice, the choice of $\mathcal{T}$ depends on human subjectivity related to the dataset. Individuals are able to incorporate their personal comprehension of semantics into the model by designing their own $\mathcal{T}$.
> > >
> > > In the experiment presented in Figure 1, we employed TPS as our data augmentation method, denoted as $\mathcal{T}$. This implies that we initially assumed that the TPS augmentation, even with its significant distortions, wouldn't alter the semantics. Thus, the resulting distorted images were **in line with our expectations**.
> > >
> > > The visual discomfort elicited by Figure 1 **solely challenges the assumption that TPS doesn't impact the semantics of CIFAR10**.
> > >
> > > ### About the distance distribution
> > >
> > > In my original rebuttal, I utilized the mathematical notation $p_{dis}$, rather than the phrase "a distance distribution", because this $p_{dis}$ is referred to the distance distribution correspond to geometric distance, e.g. Lp norm in adversarial attack context. We believe using a probabilistic model to fit this $p_{dis}$ is novel, it is not only "the practical application of a semantic distance loss to adversarial examples". We have a probabilistic perspective on this and it is not just a semantic distance loss.
> > >
> > > Thank you for bringing up VAE-GAN [1] as an example to argue that replacing a Gaussian distribution behind the L2 norm with a probabilistic generative model might not be unprecedented. While, on a high level, there may be some resemblance, it's crucial to note that VAE-GAN doesn't utilize a probabilistic generative model per se; instead, it employs a discriminator to provide a similarity metric.
> > >
> > > Probabilistic modeling is a broad concept, and crafting a specific probabilistic model doesn't inherently imply 'combining well-known techniques' as you've pointed out. Take, for example, the acclaimed Stable Diffusion [2]. It uses a diffusion model to represent the prior of the latent variable, whereas during the VAE era, this was commonly assumed to be a standard Gaussian distribution. Nonetheless, Stable Diffusion stands out as both innovative and efficacious.
> > >
> > > ### Lp distance norms should be challenged
> > > > The present work only shows experiments on digit-based datasets (MNIST & SVHN). ... Consequently, the necessity and applicability of the proposed adversarial attack are very unclear,
> > >
> > > We contend that experiments on digit-based datasets are essential. Given that the diversity of digits is markedly less than that of natural images, adversarially trained classifiers for digits tend to be more resilient to attacks. Consequently, it can be argued that targeting digit classifiers presents a greater challenge than targeting classifiers designed for natural images.
> > >
> > > > since for natural images the adversarial examples typically remain visually very close to the original inputs;
> > >
> > > 'Good enough' should not be a reason to halt further research. From our probabilistic perspective, the L2 attack can be mathematically equivalent to viewing $p_{dis}$ as a Gaussian distribution. This corresponds to the implicit assumption that 'minor Gaussian noise doesn't alter semantics.' But does this mean Gaussian noise is the optimal solution for this problem? Our findings indicate that, for digit datasets, a data-driven distribution shaped by TPS augmentation outperforms the Gaussian distribution. While CIFAR10 experiments imply that the Gaussian model might be more suitable for natural images than TPS, the pursuit of a superior augmentation should continue. Our work lays a solid foundation for this exploration.
> > >
> > > #### Reference
> > > [1] Larsen, Anders Boesen Lindbo, et al. "Autoencoding beyond pixels using a learned similarity metric." ICML 2016.
> > >
> > > [2] Rombach, Robin, et al. "High-resolution image synthesis with latent diffusion models." CVPR 2022.

---

> > > > ### Comment · Reviewer_oacq · 2023-08-15
> > > > **Re: Addressing concerns**
> > > >
> > > > I thank the authors for addressing my concerns and providing convincing arguments. I acknowledge the originality of the proposed approach in the context of adversarial examples and that the use of a probabilistic model as a potential substitute for geometric distances is indeed an interesting direction for future work. Therefore, I am willing to increase my score, but would like to use this discussion to find out to what extent.
> > > >
> > > > Currently, I still have two concerns that prevent me from increasing my score further: (i) poor visual results on CIFAR-10, and (ii) some overstatements. The latter seems to be easily remedied by watering down the text a bit, as also suggested by reviewer tkih. Regarding the first point, I recognize that the choice of $\mathcal{T}$ could be the reason for the poor visual results on CIFAR-10. Nevertheless, it is unclear to me whether a more appropriate choice of $\mathcal{T}$ could close the visual gap. Therefore, it would be valuable to provide empirical evidence for this by showing that a better choice of $\mathcal{T}$ can indeed provide adversarial examples for CIFAR-10 that are deceptive but still semantically preserving.

---

> > > > > ### Author Response · Authors · 2023-08-17
> > > > > **Updated CIFAR10 Visual Results**
> > > > >
> > > > > Thanks a lot for acknowledging the originality of our proposed approach! We are truly grateful for your patience and comprehensive review of our paper. Your suggestions and guidance have been invaluable to our progress. We would like to address your further concerns as follows:
> > > > >
> > > > > ### Improving visual result on CIFAR-10
> > > > > After further experiments, we found that given CIFAR10's data diversity, a much reduced TPS perturbation is sufficient compared to MNIST and SVHN, adjusting the standard deviation from 0.1 to 0.05. Additionally, echoing our MNIST experiment, we incorporated scaling into the data augmentation $\mathcal{T}$. Hence, our finalized $\mathcal{T}$ for CIFAR10 rests on the assumption that "minor scaling combined with slight TPS perturbation doesn't compromise the semantics of the image". We believe these updated results are a significant improvement over the previous version. Adhering to the rule, I've relayed this updated visual result to the area chair, and trust you'll receive it shortly.
> > > > >
> > > > > ### Addressing Overstatements
> > > > > We acknowledge that the initial submission of our paper had certain overstatements. Specifically, our assertion that "attacking the robust classifiers causes the image to lose its original semantics" requires further elaboration. This phenomenon primarily pertains to digit datasets like MNIST and SVHN. In the context of natural images, it's more accurate to mention that distinct colorful perturbations become apparent during attacks on robust classifiers.
> > > > >
> > > > > Additionally, instead of saying our approach "**transcends** the restriction imposed by geometric distance, opting for semantic constraints", a more precise description would be: "Our methodology leverages data-driven semantic constraints over traditional geometric distance restrictions."
> > > > >
> > > > > While we discuss in Section 3.2 that individuals can integrate their own understanding of semantics into the model by tailoring their specific $\mathcal{T}$, we will further highlight in the limitations section: "If the chosen $\mathcal{T}$ inadvertently impacts the semantics or contradicts common understanding, the generated adversarial examples may not be accepted by the majority".
> > > > >
> > > > > In addition to these points, we commit to addressing all related overstatements in the subsequent version.

---

> > > > > > ### Comment · Reviewer_oacq · 2023-08-20
> > > > > > **Re: Updated CIFAR10 Visual Results**
> > > > > >
> > > > > > I thank the authors for improving the visual results! Upon inspection, I find that the choice of $\mathcal{T}$ plays a large role; the visual results now look better. Given these additional improved visual results, I will raise my rating to "borderline accept". I would encourage the authors to include a detailed discussion of the choice of $\mathcal{T}$ for semantic preservation; perhaps the earlier results could be shown as an failure mode to make researchers and practitioners more aware of the effect of their choice of $\mathcal{T}$.

---

> > > > > > > ### Author Response · Authors · 2023-08-21
> > > > > > > **Re: Re: Updated CIFAR10 Visual Results**
> > > > > > >
> > > > > > > Thank you for your recognition! In response to your recommendation, we'll integrate a comprehensive discussion on the selection of $\mathcal{T}$ in our revised version. As previously discussed with reviewer AGWj, conducting a thorough ablation test on $\mathcal{T}$ is challenging. However, we commit to showcasing various selections of scaling and TPS within $\mathcal{T}$ (two or three scenarios for each dataset).

---

### Author Rebuttal · Authors · 2023-08-03

We extend our gratitude to all reviewers for their insightful feedback.

Attached is a PDF containing relevant figures and tables for your reference.

During the rebuttal phase, we've incorporated four additional experiments to enrich our original manuscript:

### CIFAR10 experiment

We've introduced an experiment using CIFAR10. A comparison between our method and PGD is depicted in Figure 1, with refined samples presented in Figure 2b and Figure 2c.

### Unrestricted adversarial examples

Aligning with the methodology in Song et al.'s study [1], we generated unrestricted adversarial examples using our approach. This process corresponds to formula (8) from our submission. For this purpose, we employed an energy-based model for each image class.

### Attack on certified defences

To highlight our method's efficacy, we employed it to target a certified defense [2]. Drawing a parallel to Song's approach, by overlooking the constraints of geometric distance, the theoretical boundary of such a defense becomes ineffective. This might be a contributing factor to our method's ability to sidestep certified defenses. Relevant outcomes are available in Table 1 and Table 2.

### Transferbility

The transferability of our proposed approach has also been assessed and can be viewed in Table 1 and Table 2.

### References (as linked in the attached PDF)

[1] Y. Song, R. Shu, N. Kushman, and S. Ermon. Constructing unrestricted adversarial examples with generative models. Advances in Neural Information Processing Systems, 31, 2018.

[2] E. Wong and Z. Kolter. Provable defenses against adversarial examples via the convex outer adversarial polytope. In International conference on machine learning, pages 5286–5295. PMLR, 2018.

---

### Decision · Program_Chairs · 2023-09-21

**Decision:**

Reject

**Comment:**

While the reviewers acknowledge that the work is interesting and raised their scores in response to the rebuttal. There are still remaining concerns about scalability of the approach and the limited validation on more realistic datasets, such as natural images closer to real-world resolutions. Here it will be much more challenging to train an EBM and apply the method with appropriate choices of \mathcal{T}.

This direction of work may be fruitful if it is able to scale to realistic problems, but the current paper is not able to provide evidence for this.